# Willingness to Pay for Livestock Husbandry Insurance: An Empirical Analysis of Grassland Farms in Inner Mongolia, China

**Haibin Dong [1], Saheed Olaide Jimoh [2], Yulu Hou [3] and Xiangyang Hou [1,*]**

[1] Institute of Grassland Research, Chinese Academy of Agricultural Sciences, Hohhot 010010, China; haibin3664@126.com

[2] Sustainable Environment Food and Agriculture Initiative, Lagos 104101, Nigeria; sahjin05@gmail.com

[3] Institute of Agricultural Information, Chinese Academy of Agricultural Sciences, Beijing 100081, China; houyulu@caas.cn

* Correspondence: houxy16@vip.126.com; Tel.: +86-139-1002-6776

**Abstract:** Livestock husbandry insurance (LHI) is increasingly gaining acceptance in developing countries, relative to its efficacy in mitigating the covariate risks faced by households in vulnerable agrarian communities. However, this risk-mitigating tool has received little research attention in the context of Chinese herders. The current study focused on the status, and determinants of herders' willingness to purchase LHI. We used a contingent valuation approach to collect data from 450 households across three grassland types in Inner Mongolia. Descriptive statistics and binary logistic regression models were used to analyze the collected data. We show that herders' level of awareness and acceptance of LHI are below expectations. Our results further indicated that herders with higher education level, livestock number, risk perception level, awareness, and contracted grassland area are more likely to purchase LHI. Policymakers and insurers should design programs that will educate herders on LHI while taking cognizance of other critical factors that influence households to purchase insurance. This will go a long way in scaling-up the attractiveness of LHI to herders and the agrarian community at large.

**Keywords:** risk; insurance; livestock; grassland; policy

## 1. Introduction

The agricultural economy of China is one of the largest across the globe [1]. It is characterized by the highest livestock herds worldwide, which are predominantly raised on grassland by the pastoral households. Livestock farming is highly sensitive to the vagaries of climate change (e.g., drought) that affects the pasture, livestock health, water resources, biodiversity, and the livelihood of herders that is hinged on the natural resources [2,3]. When natural pastures decline owing to climatic change or variability and in the absence of modern-risk coping measures [4], the foremost traditional approach of managing such risk by herding households include storage of forage and water for future use [5], creation of dry and wet season grazing areas, and splitting of the herd for easy management [6–8]. However, these traditional methods of risk management are less efficient due to the co-variability of weather-related challenges faced by herding households [9,10]. In this regard, livestock husbandry insurance (LHI) can be a viable market-based tool capable of guaranteeing the protection of livestock assets [4,11,12], stabilizing herders' income [13–15], and curbing the effects of uncertainty on the welfare of relatively poor households [16,17].

The implementation of LHI schemes to combat climate-related risks is increasingly gaining ground in developing countries over recent decades [4,15,18]. In China, only five agricultural insurance companies were in operation between 1949 and 2005 [19], and this number has increased to 26 firms in 2016 listing 170 insurance products for cash crops, main crops, livestock, forests, among others [1]. In 2007, the Chinese government introduced a new round of subsidized LHI [19,20], aimed at enhancing the participation of livestock farmers in insurance purchases. This is because the majority of the herders cannot afford to pay the standard premium rate set by the commercial insurers due to their low economic scale of production and capital inputs [19,21]. Furthermore, available statistics show that total premiums collected increased dramatically from USD 0.11 billion in 2006 to USD 2.26 billion in 2010, and USD 6.3 billion in 2016 covering 9.66 M ha, 45.3 M ha, and 115 M ha of cropland, respectively [1,22–24]. The scarcity of such data for LHI likely indicates that the attention of scientists and policymakers in China centers more on crop insurance programs, despite the huge market potential for LHI. To the extent that, welfare outcomes associated with catastrophic risks, herders' awareness of LHI, and their willingness to purchase such modern risk management products have received less attention in the literature. Thus, we would expect herders' knowledge of LHI and their willingness to pay for it to diverge in these contexts.

The Inner Mongolia Autonomous Region (IMAR) is one of the primary livestock keeping regions in China. The region is home to some of the most important pastoral areas in China, with 87 million ha of natural grassland accounting for 60% of its total landmass [25,26]. IMAR grasslands are famous across the world and herders in this region sustain their livelihood through livestock grazing [27]. Among the species of livestock kept in this region, sheep, cattle, and horse are of paramount importance [28,29]. This is substantiated by the large flocks of sheep, the herd of cattle, and stud of horses owned by the herding households and the existence of organized sheep and cattle market across the region and beyond [30,31]. However, livestock losses resulting from heavy snow, drought, and sandstorm are usually met by herders through local infrastructures (e.g., warm shed), borrowing from money lenders, and seldom through government loans [14,27], resulting in economic losses induced by climate-related risks. This situation points towards the need for LHI to improve the resilience of vulnerable herders through payouts in the form of income to protect households against covariate climate risks [5,18].

Several studies have been carried out on farmers' level of awareness, perception, and willingness to pay for LHI in developed and developing nations [4,5,9,13,15,32–34], but little research attention has been given to these critical aspects in the context of Chinese herders. A recent study developed a novel snow-index insurance (SII) that used the percentage grass height covered by snow as the calibration for strike (i.e., trigger for insurance payout) in eastern Inner Mongolia [27]. They conclude that the SII is superior to existing commercial mortality insurance concerning potential users' welfare. However, empirical questions that could help increase the adoption of such a new SII and other LHI products are: (1) what is the level of awareness of herders about LHI; and (2) what are the factors that influence herders' willingness to purchase LHI. Thus, a study of this type, which addresses the aforementioned research questions is relevant for policy actions in China. Results from this study may assist policymakers in understanding the factors that promote the uptake of LHI and inform plans for broader coverage in the pastoral areas. This study may also enhance future research on the methods of effectively scaling-up LHI as an adaptive mechanism to cope with climate change. Hence, the focal objective of this study is to examine the level of awareness and determinants of LHI purchase among herders in the IMAR of China.

## 2. Materials and Methods

### 2.1. The Study Site

The Inner Mongolia Autonomous Region was selected as the study area because it is one of the essential livestock production areas in China. It is located on China's northern frontier with an arid and semi-arid climate [26,29]. In 2017, the permanent population of the region was

25.28 million [25]. Vegetation distribution is dependent on the east-west gradient characterized by declining precipitation [27,31]. The grasslands in IMAR are globally recognized and play a critical role in the ecological protection of northern China. Grazing is the primary source of livelihood for households in this region. This study selected meadow steppe areas, typical steppe areas and desert steppe areas with better data representation compared to previous research in the research area.

The annual rainfall of the meadow steppe area is 350–500 mm, and the composition of herbage is rich. The perennial tufted grass and rhizomatous grass are dominant, and the main plants are Stipa baicalensis, Stipa grandis and Leymus chinensis. At the same time, due to its rich plant species, high vegetation coverage and relatively complete biodiversity preservation, it is also an excellent place for ecotourism. The annual rainfall of typical steppe area is 250–450 mm, and the rainfall is mainly concentrated in summer, and the spring is relatively dry. The main species of community construction are Stipa grandis, Stipa krylovii and Leymus chinensis. The annual rainfall of the desert steppe area is less than 200 mm, and the vegetation composition is mainly composed of perennial xerophyte grass. It is close to Hunshandake sandy land, one of the top ten sandy lands in China. It is characterized by dry climate, serious desertification and frequent sandstorms. Meanwhile, desert steppe area is also the main distribution area of fine wool sheep and Sunite sheep, and it is an important production area of animal husbandry in China.

*2.2. Data Collection*

We aim to quantify the level of awareness of herders on LHI and determine the factors that motivate them to purchase LHI. In this regard, the study was carried out by surveying a sample of 463 herder households in the meadow steppe ecosystem, typical steppe ecosystem, and desert steppe ecosystem of IMAR in 2018 (2018: the survey was conducted to collect data for 2017) (Figure 1). We employed a stratified sampling procedure to select respondents. Briefly, two counties were chosen in each grassland type. In each county, 8–12 herders were randomly selected from 2 Gacha (villages) within 3 Sumus (townships) relative to the total household number with at least 60 households interviewed in each village. Field research was conducted from September to November in 2018. Before the field survey, systematic training was conducted for the person who participated in the survey. Each questionnaire took about 1.5 h. The questionnaire collected information on household demographics, livestock and grassland, herders' cognition of LHI, perception about different forms of natural risks, and the current situation of the demand for LHI including the reasons for not purchasing insurance products if any. Semi-structured questionnaires were used in a participatory assessment method that entails face-to-face communication with the respondents during data collection. In addition, we conducted interviews with the head of the animal husbandry cooperative as well as secretaries and committee members of Gacha(s). Finally, we obtained valid feedback from 450 households representing a 97.19% return rate.

As shown in Table 1, 82.2% of the respondents were male. Most of the households were Mongolian, accounting for 89.8% of the total, and 10.2% of the Han herders. The majority of household heads were pure herders, accounting for 86.7%, and other part-time business households accounted for 13.3%. Primary school accounted for the highest proportion of education, accounting for 41.6%, followed by junior high school, accounting for 34.4%. The rest were senior high school, junior college and above, and illiterate, accounting for 15.1%, 7.1% and 1.8%, respectively.

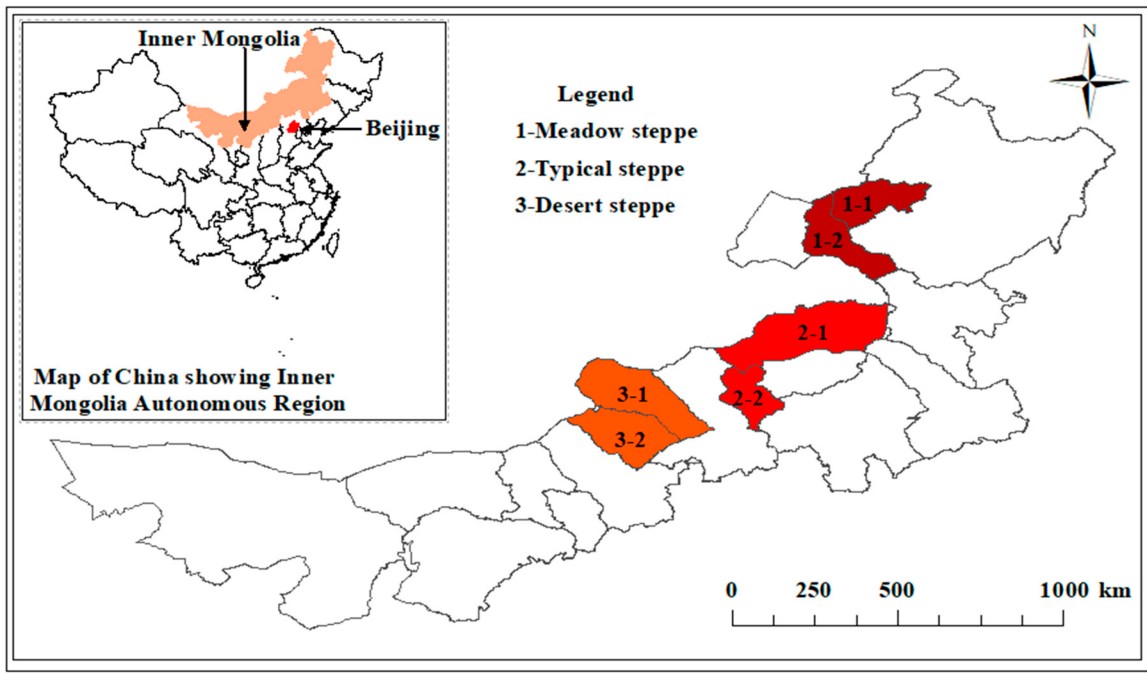

**Figure 1.** Map showing study areas across different grassland types. Note: 1-1 Chen Barag Banner; 1-2 Xin Barag Left Banner; 2-1 East Ujimqin Banner; 2-2 Xilinhot; 3-1 Sunite Left Banner; 3-2 Sunite Right Banner.

**Table 1.** Description of sample characteristics.

|  | Index | Number of Sample | Ration |
|---|---|---|---|
| Gender | Male | 370 | 82.2 |
|  | Female | 80 | 17.8 |
| Nation | Mongolia | 404 | 89.8 |
|  | Han | 46 | 10.2 |
| status | Herders | 390 | 86.7 |
|  | Concurrent business | 60 | 13.3 |
|  | Illiteracy | 8 | 1.8 |
|  | Primary school | 187 | 41.6 |
| Education level | Middle school | 155 | 34.4 |
|  | Senior school | 68 | 15.1 |
|  | College and above | 32 | 7.1 |

*2.3. Data Analyses*

2.3.1. Descriptive Statistics

We analyzed the data in SPSS version 19.0. Using descriptive analysis, we used frequencies and percentages to report the production and economic characteristics of households, the awareness level of herders about LHI, the herders' participation rate in LHI, the perception of herders about natural risks, as well as the reasons why herders do not purchase LHI. We also report the mean loss of different classes of livestock species experienced by households. To address the issue of the dimensional relationship among variables, we firstly standardize the continuous data before analysis.

2.3.2. Binary Logistic Regression Model

Our aim was to examine the significant variables that influence the willingness of herders to pay for LHI. In this sense; the response variable is a dichotomous one, represented by a dummy variable that depicts respondents' willingness to pay for insurance or not. The decision outcome was, therefore

modelled using a binary logistic regression model [4,35]. The logit model has a simple mathematical structure with less sensitivity to the distribution of sample attributes using the maximum likelihood method of estimation [15]. We generate the dependent variables by classifying herders into two groups. Group 1 comprised of herders who were willing to pay for LHI (dependent variable assigned a value of 1) and group 2 consisted of herders who were not willing to pay (dependent variable assigned a value of 0). The logit model used is:

$$P(y = 1x) = G(\beta_0 + \beta_1 X_1 + \beta_2 X_2 + \ldots + \beta_m X_m)$$

$$G(f(x)) = \exp(f(x))/[1 + \exp(f(x))]$$

where $\beta_0$ represents constant, $\beta_1, \beta_2 \ldots \beta_m$ are the regression coefficients of $X_i$ (i = 1, 2 ... m) which indicate the observed changes in the log odds of willingness to pay for LHI. An estimated positive coefficient indicates an increase in the likelihood of a herder's willingness to pay for LHI due to a unit increase in the concerned explanatory variable. $G(f(x))$ gives the odds ratio that is associated with a change in the independent variable [4,36]. The description of the explanatory variables is detailed in Table 2. The goodness of fit of the model was tested by the Hosmer Lemeshow (HL) test. The result of the HL test was not significant ($p > 0.05$), indicating that the model was a good fit for the data. The accuracy of the model prediction is 74.90%.

**Table 2.** Definition of variables considered in the binary logistic regression model.

| Variable Name | Variable Description | Mean | Std. Dev. |
|---|---|---|---|
| Age | Continuous | 0.41 | 0.17 |
| Livestock number | Continuous | 0.153 | 0.115 |
| Contracted grassland area | Continuous | 0.174 | 0.134 |
| Gender | Binary: Male = 1; 0 = Female | 1.19 | 0.39 |
| Educational level | 1 = Primary; 2 = Middle School; 3 = Senior School; 4 = College and above | 1.88 | 0.93 |
| Whether herders work with government | 1 = yes; 0 = no | 0.12 | 0.33 |
| Bank Loan | 1 = Yes; 0 = No | 0.68 | 0.47 |
| Government assistance | 1 = yes; 0 = no | 0.38 | 0.49 |
| Livestock insurance awareness | 1 = Not aware; 2 = Low level of awareness; 3 = Basic; 4 = Intermediate; 5 = Advanced | 2.22 | 1.03 |
| Risk perception level | 1 = None; 2 = Negligible; 3 = Basic; 4 = Intermediate; 5 = High | 1.85 | 1.38 |

Note: the age, gender, education level and other variables in the table are subject to the head of household of the investigated family. Government assistance mainly refers provision of hay and feed to herders at subsidized rate.

## 3. Results

### 3.1. Households' Production and Economic Characteristics

The production and economic characteristics of households presented in Table 3 show that the average livestock number owned by households was 215.34 SU in the meadow steppe, 319.47 SU in the typical steppe, and 188.03 SU in the desert steppe. The average area of grassland contracted by households across the grassland type was 365.93 hm² in the meadow steppe, 419.08 hm² in the typical steppe, and 664.16 hm² in the desert steppe, showing an increasing trend in turn. On average, the market price of livestock was relatively lower in the meadow steppe (534.11 yuan) compared to those in the typical steppe (621.06 yuan) and desert steppe (632.19 yuan), respectively. Households' access to loans was higher in the meadow steppe (214,200 yuan) and lowest in the desert steppe (98,500 yuan).

**Table 3.** Economic and animal husbandry characteristics of surveyed households.

|  | Livestock Number (SU) | Livestock Price (Yuan/SU) | Contracted Grassland (hm$^2$) | Loan (10,000 Yuan) |
|---|---|---|---|---|
| MS | 215.34 | 534.11 | 365.93 | 21.42 |
| TS | 319.47 | 621.06 | 419.08 | 13.84 |
| DS | 188.03 | 632.19 | 664.16 | 9.85 |

SU: sheep unit; GT: grassland type; MS: meadow steppe; TS: typical steppe; DS: desert steppe. Loan: household loan is reported at 10,000 yuan.

### 3.2. Herders' Awareness of Livestock Husbandry Insurance

The level of awareness of herders on the existence of LHI was 29.60%, 31.39% and 21.81% in the meadow steppe, typical steppe, and desert steppe, respectively (Table 4). The result shows that herders' awareness of LHI is generally low across the grassland types. A similar result was also obtained at the county level, where herders' level of awareness ranged from 18.95% in Sunite Left Banner in the Desert steppe to 34.48% in Xilinhot City in the typical steppe (Table 5). This points at the need to sensitize herders on LHI across the study areas.

**Table 4.** Herders level of awareness of livestock husbandry insurance across grassland type.

|  | Meadow Steppe | Typical Steppe | Dessert Steppe |
|---|---|---|---|
| Yes | 29.60 | 31.39 | 21.81 |
| No | 70.40 | 68.61 | 78.19 |

**Table 5.** Herders level of awareness of livestock husbandry insurance across the surveyed counties.

|  | Meadow Steppe | | Typical Steppe | | Dessert Steppe | |
|---|---|---|---|---|---|---|
|  | Chenbarhu Banner | Xin Barag Left Banner | Xilinhot City | East Ujimqin Banner | Sunite Right Banner | Sunite Left Banner |
| Yes | 30.16 | 29.03 | 34.48 | 29.11 | 24.73 | 18.95 |
| No | 69.84 | 70.97 | 65.52 | 70.89 | 75.27 | 81.05 |

### 3.3. Distribution of Herders' Willingness to Purchase Livestock Husbandry Insurance

In the meadow steppe, only 24.80% of respondents were willing to purchase LHI in contrast to 75.20% of respondents who were not willing to pay for insurance (Table 6). For typical and desert steppe, 26.28% and 20.21% of respondents, respectively, were willing to purchase LHI, while the remaining 73.72% and 79.79% were not interested in paying for insurance products. At the county level, herders who were willing to purchase LHI were not more than 15.33%, which was recorded in East Ujimqin Banner in the typical steppe (Table 7). Taken altogether, the results indicated the low attention and demand for LHI by herders in the study areas, mainly because the majority of the herders are not aware of the existence of LHI.

**Table 6.** Household response to willingness to purchase livestock husbandry insurance across grassland type.

|  | Meadow Steppe | Typical Steppe | Dessert Steppe |
|---|---|---|---|
| Purchase | 24.80 | 26.28 | 20.21 |
| Not purchase | 75.20 | 73.72 | 79.79 |

**Table 7.** Household response to willingness to purchase livestock husbandry insurance across the surveyed counties.

| | Meadow Steppe | | Typical Steppe | | Dessert Steppe | |
|---|---|---|---|---|---|---|
| | Chenbarhu Banner | Xin Barag Left Banner | Xilinhot City | East Ujimqin Banner | Sunite Right Banner | Sunite Left Banner |
| Purchase | 14.40 | 10.40 | 10.95 | 15.33 | 10.64 | 9.57 |
| Not purchase | 85.60 | 89.60 | 89.05 | 84.67 | 89.36 | 90.43 |

*3.4. Reasons for Non-Willingness to Purchase Livestock Husbandry Insurance*

We asked herders who were not willing to purchase LHI about the possible reasons that led to their decision. Among the reasons provided, the highest number of households responded with "others" which was 49.60% in the meadow steppe, 72.99% in the typical steppe, and 71.10% in the desert steppe, respectively (Table 8). These other reasons include having less livestock than the standard quantity that qualifies for insurance, and the proportion of animals herders can afford to insure from their herd. The other two most important factors responsible for herders' decisions not to purchase LHI across the grassland type are "inadequate knowledge" which indicates a lack of understanding and "no need of insurance" (obtained through interviews). Further, inadequate knowledge shows that herders do not have a good understanding of LHI, and therefore need proper sensitization to become familiar with its modus operandi. Herders' perception that they do not need insurance could be linked to several factors such as low level of awareness, lack of access to extension outreach programs, and the medium employed by insurance companies for advertisement. This suggests the need for a better insurance awareness campaign among herders [32].

**Table 8.** Reasons for non-willingness to purchase livestock husbandry insurance.

| | No Need of Insurance | Inadequate Knowledge | High Benefits | Low Coverage | Poor Service Quality | Others |
|---|---|---|---|---|---|---|
| MS | 12.80 | 30.40 | 7.20 | 3.20 | 1.60 | 49.60 |
| TS | 10.20 | 13.10 | 0.70 | 0.00 | 2.90 | 72.99 |
| DS | 3.70 | 23.50 | 0.00 | 0.00 | 1.60 | 71.10 |

GT: grassland type; MS: meadow steppe; TS: typical steppe; DS: desert steppe.

*3.5. Herders' Perception of Natural Risks*

There is variation in the level of herders' concern about natural disasters such as drought, sandstorm, snow disaster, and fire, which directly affects livestock production and the livelihood of herders (Table 9). Specifically, drought is the most threatening natural risk to herders across the grassland types in the order desert steppe > typical steppe > meadow steppe. Drought is accompanied by multidimensional effects such as decreased grassland productivity, poor livestock condition, and the impairment of households' welfare. In addition, herders in the meadow and typical steppe are also concerned about snowstorms, while those in the desert steppe also nurture fear about sandstorms. Our results indicate that households are faced with single or compound natural risks during the annual production cycle. This suggests that LHI is a promising risk management tool that could help mitigate the effects of natural risks to stabilize animal husbandry production and herders' income in the study area. Therefore, LHI may be needed by herders in this area to combat natural risk. Besides, this risk mitigating tool has a considerable market potential in this region.

**Table 9.** Herders' perception of natural risks.

|  | Drought | | Sand Storm | | Snowstorm | | Fire | | Others | |
|---|---|---|---|---|---|---|---|---|---|---|
|  | **Worry** | **Most Worried** | **Worry** | **Most Worried** | **Worry** | **Most Worried** | **Worry** | **Most Worried** | **Worry** | **Most Worried** |
| MS | 94.40 | 81.60 | 4.00 | 1.60 | 74.40 | 11.20 | 3.20 | 2.40 | 0.80 | 0 |
| TS | 98.54 | 85.40 | 23.36 | 3.65 | 42.26 | 10.95 | 4.38 | 0 | 14.59 | 0 |
| DS | 99.47 | 95.21 | 87.77 | 4.79 | 3.72 | 0 | 1.60 | 0 | 1.06 | 0 |

GT: grassland type; MS: meadow steppe; TS: typical steppe; DS: desert steppe. Note: other risks mainly refer to the infrequent floods, windstorms, rats and insects in the study area.

*3.6. Herders' Perception of Losses Due to Catastrophic Events*

Herders' perception of losses due to catastrophic events is shown in Table 10. The result shows a varying degree of livestock losses by households across the grassland type due to factors such as climate disaster, epidemic disease, and mining pollution. The production losses faced by herders calls for the need to reduce livestock losses as a result of catastrophic events and improve their income and welfare [13,36]. The most severe losses are those of sheep, with resulting economic impact on the routine production and livelihood of herders. For example, in the meadow steppe, the economic value of the adult sheep lost is estimated at 8600 yuan using the average selling price in 2017. This will have a significant effect on herders' net livestock income. The results also revealed that the annual death rate of adult sheep and cattle across the three grassland types were higher than those of their corresponding lambs, calves, and ponies. Again, on an economic scale, the former is accompanied by a higher loss to herders than the latter. Notably, herders in the meadow steppe are more vulnerable to higher livestock losses. Therefore, there is a need for increased awareness of LHI in the research area. This will have a significant practical implication in stabilizing the income of herders, promoting sustainable livelihood, and driving a stable development of the regional economy.

**Table 10.** Summary of household livestock losses due to catastrophic events.

|  | Number of Deaths | | | Number of Lambing Loss | | |
|---|---|---|---|---|---|---|
|  | **Sheep** | **Cattle** | **Horse** | **Lambs** | **Calves** | **Ponies** |
| MS | 10.83 | 1.76 | 0.42 | 5.26 | 0.50 | 1.45 |
| TS | 5.07 | 1.05 | 0.12 | 3.98 | 0.30 | 0.13 |
| DS | 6.55 | 0.25 | 0.05 | 3.99 | 0.20 | 0.14 |
| Total | 22.45 | 3.06 | 0.59 | 13.23 | 1.00 | 1.72 |

GT: grassland type; MS: meadow steppe; TS: typical steppe; DS: desert steppe. Sheep, cattle, and horse refers to mature animals.

*3.7. Determinants of Herders' Willingness to Purchase Livestock Husbandry Insurance*

We fitted the binary logistic regression model in SPSS 19.0. Our result revealed that the likelihood ratio chi-square of the model is 51.627 with degrees of freedom 10 at $p < 0.001$. The Hosmer Lemeshow (HL) test returned a value of 18.937, with a p value of 0.11, indicating that the overall goodness of fit of the model is good. The Cox and Snell R square is 10.8% and Nagelkerke R Square is 15.8%. These test statistics explains the variations caused by the explanatory variables on the herders' willingness to pay for LHI. Results of the fitted logit model to identify the factors influencing the willingness of herders to pay for LHI are displayed in Table 11.

**Table 11.** Logit estimates of the factors influencing herders' willingness to pay for livestock husbandry insurance.

| Independent Variables | B | S.E | Exp |
|---|---|---|---|
| Age | 0.532 | 0.727 | 1.702 |
| Gender | −0.126 | 0.298 | 0.881 |
| Educational level | 0.371 *** | 0.137 | 1.449 |
| Livestock Number | 3.101 *** | 1.020 | 22.211 |
| Contracted Grassland | 2.019 ** | 0.886 | 7.529 |
| Whether herders work with government | 0.153 | 0.350 | 1.165 |
| Livestock insurance awareness | 0.246 ** | 0.112 | 1.297 |
| Bank Loan | 0.050 | 0.248 | 1.051 |
| Risk perception level | 0.318 *** | 0.079 | 1.375 |
| Government Assistance | 0.161 | 0.246 | 1.175 |
| constant | −3.972 *** | 0.760 | 0.019 |
| Comprehensive test of model coefficient Sig = 0.000 | | −2Log likelihood = 470.297 | |
| Cox–Snell R Square = 0.108 | | Nagelkerke R Square = 0.158 | |

Note: *** and ** are statistically significant at 1% and 5%, respectively. Government assistance mainly refers to provision of hay and feed to herders at a subsidized rate.

As can be seen from Table 11, the variables that have significant impact on herders' willingness to purchase LHI are education level, livestock number, contracted grassland, livestock insurance awareness and risk perception level. Specifically, the variable of education level has a significant positive impact on herders' willingness to purchase LHI at the level of 1%. Moreover, for every increase in the level of education, the herders' willingness to purchase LHI will increase by 0.371 units. The livestock number and risk perception level also significantly affect herders' willingness to purchase LHI at the 1% level, and both are positive effects. Besides, for every increase in the livestock number and risk perception level, the herders' willingness to purchase LHI will increase by 3.101 units and 0.318 units, respectively. The contracted grassland variable significantly affects the willingness of herders to purchase LHI at the 5% level, and has a positive effect. In addition, for each increase in the contracted grassland, the willingness of herders to purchase LHI will increase by 2.019 units. The variable of livestock insurance awareness has a significant positive impact on the willingness of herders to purchase LHI at the level of 1%. Moreover, for each increase in the livestock insurance awareness, the willingness of herders to purchase LHI will increase by 0.246 units.

Actually, education level, livestock number, contracted grassland, livestock insurance awareness and risk perception level are not only the main factors affecting the willingness of herders to purchase LHI, but also the key aspects to ensure the sustainable development of herders' livelihood.

## 4. Discussion

Our empirical analysis shows that herders' willingness to purchase LHI is not affected by age, gender, whether herders work in the Gacha or not, government assistance, and access to bank loan. This contradicts earlier reports that have shown positive [37,38] and negative [9,14] effects of age and gender on farmers willingness to participate in insurance. However, Dessart et al. (2019) argued that age is not a behavioral factor that can influence farmers' decisions on agricultural policies. Other studies have also reported the non-influence of gender on the uptake of crop [37] and livestock insurance [4] in Kenya and India.

The result of the logistic regression showed that the level of formal household education positively influences the willingness of herders to purchase LHI. The odds ratio in favor of herders' willingness to purchase LHI increases by 1.45 for every year of increased education. Our result corroborates the earlier reports [9,15,32] that household education had a positive impact on farmers' decision to adopt crop and livestock insurance. This can be attributed to three possible reasons: (1) education assists farmers/herders to understand the importance of insurance as a useful tool in mitigating risk in their livestock production [4,39]; (2) education improves the knowledge of herders about the consequences

of climate change and to see reasons to purchase LHI to minimize its possible impact [5,40]; and (3) education enhances the adoption of livestock insurance by broadening the thinking of herders to enhance sound decision making related to risk management [9,33].

As the number of herders' livestock increases, so does the odds ratio in favor of purchasing LHI. Hence, an increase in livestock number necessitates the need for insurance to mitigate livestock losses caused by the covariate risk faced by herding households [13,32]. Similarly, other studies also reported a positive correlation between herd size and household's willingness to pay for index-based livestock insurance (IBLI) in Ethiopia [5] and Namibia [32]. Another advantage of LHI in this regard is its potential to reasonably decline distressful herd off-take before or after catastrophic risk which could help to maintain both a household's economic growth and the grassland ecosystem health [8,18]. However, it is worth noting that sustainable grazing is a function of grassland sensitivity to defoliation by grazing animals [18]. The size of landholding is associated with livestock productivity, higher income, and the ability to pay for agricultural innovations such as insurance [4,5]. In this study, we observed a positive relationship between the area of grassland contracted by herders and their ensuing willingness to purchase LHI. This implies that the probability of purchasing LHI increases with an increase in grassland contracted area [14,41]. Similar results have been reported in France and Italy, where farm size influenced farmers' decision to participate in crop insurance [42].

From the econometric analysis, we found that the odds of herders purchasing LHI increases by 1.30 with a unit increase in the level of awareness. This lends support to the results from Nigeria [9], Kenya [33,37], and India [4] that awareness positively influenced farmers and herders' adoption of insurance. Our results suggest the need for insurers, policymakers, and research institutes within the study area to design programs that will focus on educating herders on the potential benefits of insurance (e.g., climate change adaptation; reduced distressful herd off-take) to improve herders' knowledge of insurance and subsequent uptake of it. For example, in Ethiopia, increased awareness about IBLI successfully increased livestock number insured and the rate of households' participation from 2012 to 2017 [5]. In this sense, LHI is capable of improving household welfare and the health of the common property, with an attendant environmental benefit such as economic development and adaptation [43].

According to [44], risk perception implies an individual's assessment of the potential effect of risk in a particular situation. The econometrics results further showed that herders' risk perception level is an essential factor in determining their willingness to purchase LHI. With a unit increase in the risk perception level of herders, the odds of purchasing LHI increases by 1.38. This is in agreement with the extant literature as well as our a priori expectation [9,27,36,43]. Although the perception of how much risk is mitigated by insurance may vary among individual herders, however, LHI remains a holistic risk management framework that can improve the resilience of vulnerable herders, promote ecological sustainability, and contribute to agricultural modernization [5,11,18,45–47]. To this end, there is a need for further research in the study areas to determine the range of premiums herders are willing to pay for insuring their livestock and the type of insurance products they are likely to patronize. This would help to couple policy insights with designing an acceptable insurance scheme that will boost herders' confidence to give credence to insurance as a necessary and useful risk management structure.

## 5. Conclusions

This study was designed to understand the status of livestock husbandry insurance (LHI) across the meadow steppe, typical steppe, and desert steppe of Inner Mongolia, and investigate the critical factors influencing herders' willingness to pay for it. Our analysis has shown that herders' level of awareness and acceptance of LHI are below expectations. Besides other reasons such as the possession of small livestock assets that impedes participation in LHI, a majority of herders indicated that the knowledge at their disposal on LHI is inadequate to decide on the purchase of it. This demonstrates that herders need to know more about LHI to make decisive decisions about it. The result also indicated that households' livestock losses due to catastrophic events were higher for adult sheep than

other livestock species and classes, with a perceived economic effect on household livestock income. This suggests the potential need for LHI to cushion the economic shock that can be anticipated from such losses when catastrophic events occur.

The binary logistic regression model results showed that the willingness to purchase livestock insurance is positively influenced by education level, livestock number, risk perception level, awareness, and contracted grassland area. The result suggests the need for policymakers and insurers to design programs that will educate herders on risk management tools (e.g., financial literacy) to improve herders' awareness of LHI and help them make an informed decision when purchasing insurance products. In addition, government and other stakeholders such as research institutes should make a concerted effort towards policies and outreach programs that will enhance the factors that influence willingness to pay for insurance as found in this study. Insurance products and programs designed to communicate to the herders should be flexible enough to meet the target audience's need concerning product design, channels of information delivery, etc. The findings from this study narrow down the knowledge gap related to the promotion of LHI uptake in the study area and recommend scaling-up awareness about livestock insurance to enhance its acceptance by herders.

This article mainly studies the willingness of herders to purchase animal husbandry insurance. In future research, we will further quantify the effect of purchasing animal husbandry insurance on the protection of herders' livestock production; our aim will be to make new and greater contributions for the implementation of animal husbandry insurance in grassland pastoral areas and protect the livelihood of herders.

**Author Contributions:** Conceptualization, H.D.; Methodology, H.D.; Software, H.D.; Validation, H.D. and S.O.J.; Formal Analysis, H.D. and S.O.J.; Investigation, H.D. and S.O.J. and Y.H.; Resources, X.H.; Data Curation, H.D. and Y.H.; Writing—Original Draft Preparation, H.D.; Writing—Review and Editing, H.D. and S.O.J.; Visualization, X.H.; Supervision, X.H.; Project Administration, X.H.; and Funding Acquisition, X.H. All authors have read and agreed to the published version of the manuscript.

**Funding:** This research was supported by National Natural Science Foundation of China (No. 71774162), Soft Science Project of State Forestry and Grassland Administration of China (No. 2019131042) and Basic scientific research business expenses of central public welfare scientific research institutes in China (No. 1610332018011).

**Acknowledgments:** We thank our colleagues who assisted us during the field survey and the herders who patiently answered our questions during this study.

**Conflicts of Interest:** The authors declare no conflict of interest.

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
