# Peer review of "Willingness to Pay for Livestock Husbandry Insurance: An Empirical Analysis of Grassland Farms in Inner Mongolia, China"

_sustainability, doi:10.3390/su12187331_

Round 1
Reviewer 1 Report
Thank you for resubmitting the manuscript! It reads much better and I recommend only minor clarifications.
Line 62: correct word "stud" - to study.
Lines 76-77. Suggest to rephrase - the question - what is the level of awareness does not increase the adoption.. - but the question could help increase the understanding of the LHI adoption rate.
Line 82. suggest : as an adaptive mechanism to COPE with climate change.
Line 95. Explain what is meant by better data representation - previous research - or availability of data from government agencies. (specify which).
Line 111. Motivate to purchase THE same / or to purchase LHI.
Line 118: Systematic training was conducted for the researchers. NOT clear. Is not the authors of this paper the researchers?
Line 126. 82,2 % of the households were male. What is meant by this sentence? Does it mean that all members of the household were male?
Line 187: A bit confusing use of respondents and herders in the same sentence. Are not referred to respondents also herders in this context?
Line 345. I suggest to replace "wish" with "need".
Reviewer 2 Report
I would suggest that you rewrite section 3.7 into "Discussion".
Author Response
Please see the attachment.

This manuscript is a resubmission of an earlier submission. The following is a list of the peer review reports and author responses from that submission.
Round 1
Reviewer 1 Report
Introduction
First paragraph – rework language. Several too long sentences. Each sentence refers to too many concepts – reduces the easiness of reading. An example is line 32 - 36.
In line 38, “Livestock husbandry insurance” the main topic of the paper is referred to for the first time. It is stated that "it is a viable market-based tool.. It is recommended to revise to; … that it “can” be a viable marked based tool. It is hardly in all types circumstances, "a vible tool". The authors could as an alternative refer to other studies /authors which show that it “is” a a viable market.
It is recommended to describe more clearly what this LHI involves. It is briefly mentioned that it is subsidized. But the reader is still largely unaware of what exactly the LHI mean, what time of cost, contract with whom etc.
Line 49 It is stated that the demand for LHI, “remains very low relative to crop insurance purchase by farmers”. Does this refers to studies of farmers in the same area? If not, it is difficult to see how it can be comparable? It is thought that this perhaps does not refer to the case study area, as the case study area is “primary livestock keeping region in China”
The authors explain why the region was selected as the study area. It is recommended to also explain why the division into “meadow steppe”, typical steppe, and desert steppe”. There are surely some typical characteristics associated with each of these, which are relevant for the selection, data collection.
Data collection:
The authors need to inform about , what type of questions were included in the semi-structure questionnaire. What is the rationality for asking these questions? Any pre interviews to make sure that the relevant questions were asked? What theory / theories were important when selecting questions for interview? And how long for each interview, and when was the data collected.
It is recommended to add a section on the study area. Too little information about the different case study area, what are typical characteristics and history in the different “steppe”. This may very well contribute to an explanation of the different willingness to pay for the LHI.
Results and discussion
It may be recommended to separate results and the discussion section to clarify what are results, and what are discussion.
The paper in general provides several statements which are not adequately accounted for. Example, line 154 and 155, “Market price depict the situation in 2017”. What situation is being depicted? Please explain.
Lines, 157 – 158 “Households’ access to loan was higher in the meadow steppe and lowest in the desert steppe”. Why is this?
Lines 166-167: it is referred to a study at county level? Is this the same study? Again it is needed to better give an overview of what are case study areas? A table to show characteristics, area etc. is recommended.
Lines 172: “Our result shows a similar trend..” What study do the authors make a reference / comparison to?
Lines 191-193 It is here referred to the result that herders point to “inadequate knowledge” as an important factor. The authors explain, understand this as, “lack of understanding, and “no need for insurance”. Is this supported by interviews, or in any other way supported ? Otherwise, I am tempted to indicate that this is merely speculation from the authors. “Inadequate knowledge” – could refer to “inadequate knowledge of the costs, of the contract, or several other issues.
Lines 212-213, The authors write “ the herders’ in this area need LHI to combat natural risk.. “ . It recommended to rewrite this sentence, as it gives an impression that the author is an authority. Furthermore, it is not sufficiently accounted for.
Reviewer 2 Report
The economics of livestock insurance is interesting from the perspective of farmers, insurers and government. In particular, know-how how to design livestock husbandry insurance is the challenge for insurers (also public administration in the context of PPP). Pastoral rural economics (Inner Mongolia is a good case) need a detailed toolbox of risk management tools in order to
Streghts
The context of Chinese agriculture is well highlighted (Introduction) The empirical strong evidence (primary data) with a nice visualisation The correct empirical approach that is typical for assesment WTP(willingess to pay)
Weaks
The lack of the road-map paragraph describing a general design of the article The lack of data on the context - livestock husbandry sector in China (e.g. production, efficiency etc.) and comments on "governance assistance" (used as "variable") The detailed statistical description of the sample is missing (only mean and SD The short concluding remarks (related to research limitation and avenues for future research) The nexus between insurance policies and the rest of risk management tools is not described What about methodological approaches for WTP in crop production? The lack of a synthesis (review) of aforesaid approaches is one of shortcomings The aspect of novelties/innovations for livestock insurance products should be strengthened in discussion/concluding remarks
Detailed remarks
Please consider detailed 3-4 hypotheses - research empirical questions seem to be very general. In my opinion, there are some missing or only partially developed parts (I mentioned in "Weaks"), e.g road map paragraph in "Introduction", data on the context - livestock husbandry sector in China, the detailed statistical description of the sample. Concluding remarks should be more detailed and reconsidered, incl. research limitation and avenues for future research, incl. the aspect of novelties/innovations for livestock insurance products.
Reviewer 3 Report
The paper addresses an important problem of risk management and the role of agricultural insurance in animal husbandry. The practical use of insurance in agriculture is a difficult topic due to the systemic nature of risk, information asymmetries and moral hazard. In my opinion, these issues should be better described in the theoretical part of the paper, as they justify subsidizing of agricultural insurance in many countries.
Taking into account the profile of the journal, in addition, it should be discussed why the problem of risk is important for agricultural sustainability in the theoretical part of the paper.
However, my main doubt is whether searching for the determinants of "willingness to pay for insurance" is justified in the case of farmers who have never heard of this insurance ? I think we have 2 separate research groups here - aware and unaware farmers. One can hypothesize that in the group of unaware farmers the basic determinant of the lack of willingness to pay is the lack of awareness (and knowledge). However, this factor is irrelevant for aware farmers and in this case searching others determinants seems to be justified . So I ask the question whether in this context two separate logistic regression models should be prepared (for the aware and the unaware) - but maybe I am wrong, it's just something to think about.
I have also a few minor comments:
in my opinion in section 2.3.1. - a table with detailed farmers characteristics divided into 2 groups: "willing to pay" and "not willing to pay" should be added I think section 2.3.2 is redundant - these are just 2 sentences line 142:: The result was not significant (p> 0.05), indicating that the model was a good fit for the data. "- Are you sure that the lack of significance indicates a good fit of the model ??? line 150: "The socio-economic characteristics of households presented in Table 2" - what parameters of "socio" characteristics are in Table 2 - in my opinion it is only production (and economic ) characteristics in table 2, column 2 - I guess the correct unit of measure is SU / farm? all tables should have the source provided The rating scale in Table 8 is incomprehensible for me, there are 2 categories: "worry" and "most worried" - what about others? In the case of "draught" the sum of "worry" and "most worried" exceeds 100% - how do you understand it? In addition, the horizontal arrows in the tables (indicating the "%" sign) are probably superfluous (??). Table 10 should probably be placed in the section 3.7 and not in 3.6 Considering that there are many different motifs and tables in the text, it would be good to add an additional graph with the conceptual framework at the beginning of the paper (it would show the mutual relations of various parts of the study).